# Cancer Stem Cells and Their Possible Implications in Cervical Cancer: A Short Review

**DOI:** 10.3390/ijms23095167

**Published:** 2022-05-05

**Authors:** Riccardo Di Fiore, Sherif Suleiman, Rosa Drago-Ferrante, Yashwanth Subbannayya, Francesca Pentimalli, Antonio Giordano, Jean Calleja-Agius

**Affiliations:** 1Department of Anatomy, Faculty of Medicine and Surgery, University of Malta, MSD 2080 Msida, Malta; sherif.s.suleiman@um.edu.mt; 2Sbarro Institute for Cancer Research and Molecular Medicine, Center for Biotechnology, College of Science and Technology, Temple University, Philadelphia, PA 19122, USA; president@shro.org; 3BioDNA Laboratories, Malta Life Sciences Park, SGN 3000 San Gwann, Malta; rosa.dragoferrante@biodna.net; 4Centre of Molecular Inflammation Research (CEMIR), Department of Clinical and Molecular Medicine (IKOM), Norwegian University of Science and Technology, 7491 Trondheim, Norway; yashwanth.subbannayya@ntnu.no; 5Department of Medicine and Surgery, LUM University “Giuseppe DeGennaro”, 70010 Casamassima, Italy; pentimalli@lum.it; 6Department of Medical Biotechnologies, University of Siena, 53100 Siena, Italy

**Keywords:** cervical cancer, cancer stem cells, drug resistance, radio-resistance, EMT, quiescence, epigenetic, targeted therapy, delivery systems

## Abstract

Cervical cancer (CC) is the fourth most common type of gynecological malignancy affecting females worldwide. Most CC cases are linked to infection with high-risk human papillomaviruses (HPV). There has been a significant decrease in the incidence and death rate of CC due to effective cervical Pap smear screening and administration of vaccines. However, this is not equally available throughout different societies. The prognosis of patients with advanced or recurrent CC is particularly poor, with a one-year relative survival rate of a maximum of 20%. Increasing evidence suggests that cancer stem cells (CSCs) may play an important role in CC tumorigenesis, metastasis, relapse, and chemo/radio-resistance, thus representing potential targets for a better therapeutic outcome. CSCs are a small subpopulation of tumor cells with self-renewing ability, which can differentiate into heterogeneous tumor cell types, thus creating a progeny of cells constituting the bulk of tumors. Since cervical CSCs (CCSC) are difficult to identify, this has led to the search for different markers (e.g., ABCG2, *ITGA6* (CD49f), *PROM1* (CD133), *KRT17* (CK17), MSI1, *POU5F1* (OCT4), and SOX2). Promising therapeutic strategies targeting CSC-signaling pathways and the CSC niche are currently under development. Here, we provide an overview of CC and CCSCs, describing the phenotypes of CCSCs and the potential of targeting CCSCs in the management of CC.

## 1. Introduction

Cervical cancer (CC) is the fourth most common cancer in women worldwide. The majority of cases of CC are linked to infection with human papillomaviruses (HPV) [1]. Although most infections with HPV resolve spontaneously and are asymptomatic, persistent infection with the high-risk types can cause CC in women. Several lifestyle factors, such as multiple sexual partners and smoking, enhance the progression of high-risk HPV infection to CC [2].

In the last couple of decades, there has been a significant improvement in the prevention of CC thanks to effective Pap smear screening and administration of vaccines. This has led to a lower incidence and mortality in high resource countries. For example, it is estimated that CC in Australia will be nearly eliminated by the end of this decade [3]. However, in low-income countries this is unlikely to be achieved even by the end of the century. Once there is a global scale-up of similar public health measures, especially in low-resource countries, CC may come to be considered a ‘rare’ disease in these societies in the future [1]. However, due to poor access to preventive health care, CC is still to date, the fourth most common type of gynecological malignancy affecting females worldwide [4]. In 2018, CC accounted for 6.6% and 7.5% of female tumor morbidity and mortality [4]. The prognosis of patients with advanced and/or recurrent CC is particularly poor, with a one-year relative survival rate of a maximum of 20% [5].

If it is detected early and managed effectively, CC can be successfully treated using surgery and/or radiotherapy. For micro-invasive carcinomas (stage IA1) and small-volume macroscopic disease (IB1 and IIA1), patients are advised to undergo conization and hysterectomy [6]. More advanced CC cases are typically treated using a combination of radiotherapy and chemotherapy. For cases in the locally advanced stage, radiation together with cisplatin-based chemotherapy is the primary therapeutic option [7]. In addition, cell cycle-specific drugs, including vincristine, paclitaxel, 5-fluorouracil, and gemcitabine, have radiosensitization capabilities or synergize the cytotoxic effects of platinum drugs. However, in advanced-stage disease, systemic chemotherapy has a limited effect. Therefore, novel agents are urgently needed for a better therapeutic result.

With the development of molecular biology and omics technology, breakthroughs have been achieved in targeted therapy research, including immune checkpoint inhibitors, anti-angiogenesis agents, poly (ADP-ribose) polymerase (PARP) inhibitors, and other potential treatments for CC [8]. Targeted gene delivery therapy is another promising approach leading to the development of multiple strategies, ranging from immune system potentiation, altered gene restoration, oncolytic virotherapy to the use of nanotechnology paving the way to designing improved and enhanced gene delivery systems. Multiple targeted gene delivery systems have been developed to improve tumor targeting and minimize toxicity in normal tissue with encouraging pre-clinical results [9]. However, the clinical translation to humans is still lagging mainly due to the lack of efficient vectors. [9]. In addition, several novel compounds derived from microorganisms or plants have been shown to have prominent anti-cancer activity through changes in the apoptotic balance in CC [10].

Another crucial factor affecting the management and prognosis in patients with CC is the presence of CC stem cells (CCSCs), which represent a small subpopulation of tumor cells with a high potential for self-renewal, a multilineage differentiation, tumorigenicity, and a slow-cycling capacity [11,12]. Since CCSCs are more resistant to conventional treatments, such as different chemotherapy and radiotherapy regimens [13], studies are being carried out to target these cells. Therapeutic targeting of CCSC has the potential to reduce the tumor burden by preventing the generation of new CC clones, and therefore, not only prevents resistance to conventional therapies but also limits distant metastasis and relapse.

This review aims to provide an update on CC and CCSCs, including a description of CCSCs’ phenotypes and an outline of the potential of targeting CCSCs in the treatment of CC.

## 2. Cancer Stem Cells in Cervical Cancer

The “clonal evolution” theory of carcinogenesis suggests that CC arises due to a mechanism of loss of control, leading to unlimited and unharnessed cellular proliferation in cells of clonal origin with similar molecular characteristics [14]. However, there is increasing evidence of intratumoral heterogeneity in CC.

One explanation for the heterogeneity in CC is the existence of CCSCs. These slow-cycling CCSCs reside in the niche areas of the tumors and are capable of initiating and maintaining neoplastic growth as well as leading to distant metastasis [15]. These distinct tumor cell populations exhibit different molecular and phenotypic characteristics associated with a poor response to chemo- and/or radiotherapy and increased risk of lymph node metastasis and pelvic recurrence in CC [16]. Given that CSCs typically undergo asymmetric division, histological examination of CC tissue exhibits a heterogeneous population of diversely differentiated carcinoma cells. Another contributing factor to the tumor heterogeneity is due to the ability of CSCs to transdifferentiate into vascular endothelial cells and other tumor-associated stromal cells [17].

Most CCs are linked to infections caused by high-risk strains of HPVs (hrHPVs) [18]. While the post-infection microenvironment facilitating viral persistence is becoming increasingly recognized for CC malignant progression, being infected with HPV during one’s lifetime does not necessarily inevitably lead to neoplastic transformation [19]. In fact, the majority of hrHPVs infections are cleared spontaneously, with only around 10% to 15% persisting and eventually leading to the progression of precancerous cervical intraepithelial neoplasia (CIN) to invasive CC [18]. HPV-scoring systems, weighing resultant gene alterations, are being developed as prediction tools in predicting the prognosis of CC by evaluating individual HPV infection status and any subsequent genetic modification [20].

Approximately 90% of CIN3 and CC arise within the squamo-columnar junction, a transition area between the exocervix and endocervix [21]. These specific squamo-columnar junction cells exhibit junction-specific markers which are similar to those expressed in carcinogenic HPV-associated CINs and carcinomas. These include both squamous cell carcinomas and adenocarcinomas, indicating that multiple cervix malignancy subtypes are derived from the squamo-columnar junction cells [22,23]. It has been hypothesized that the squamo-columnar junction may harbor stem-like cells, which, in the presence of the persistent infection with carcinogenic HPV, increase the risk of developing CC [13].

Although some markers for CSCs have been identified, there is no collection of universal biomarkers for specifically identifying and isolating CSCs [24]. This is mainly due to the heterogeneity of CSCs at both the intratumor- and intertumor-type levels. Therefore, in order to isolate CSCs within a particular tumor site and across several tumor sites, a variety of cell surface and functional markers need to be used [25]. This also applies to CCSC markers where there are variations from tumor to tumor, and therefore, CC cells expressing a single stem cell marker do not always qualify as CCSCs. However, novel markers for CCSCs are being identified and further investigated in the hope of enabling diverse therapeutic options to cure CC [13,15]. A brief list of studies on CCSC phenotypes is provided in Table 1 [26,27,28,29,30,31,32,33,34,35,36,37,38,39,40,41,42,43,44]. However, it is still not clear whether the difference in the stemness expression profile will translate into a clinically relevant difference in CSC phenotype and successful outcome when implementing this therapy.

## 3. Cancer Stem Cells and Therapeutic Implication

There are very limited data available that validate and support the clinical diagnostic value of CCSC biomarkers. The current understanding of these biomarkers suggests that most of them indicate progression of lesions that are already initiated [15]. However, these markers may not be very sensitive to identify all initiated lesions. Several biomarkers, especially when used in combination, have been identified for the screening of CC [13,15].

Surgery, chemotherapy (particularly cisplatin), and radiotherapy have improved the overall survival of patients with CC. However, the presence of CCSCs that are resistant to chemo- and radiotherapy leads to disease relapse and a reduction in overall survival [13]. CCSCs can develop resistance to standard treatments via different mechanisms (Figure 1), which are further described in detail below. Owing to their tumorigenicity, CSCs may be the route of cervical carcinogenesis, leading to distant metastasis. Therefore, therapeutic management specifically targeting CSCs is a potential tool for preventing chemo/radio-resistance and decreasing the risk of distant metastasis, tumor relapse, and the generation of secondary tumors, thereby increasing the chances of CC patient survival [45]. The identification of CCSC and a deeper understanding of their microenvironment will enable their specific pharmacological targeting [13].

## 4. The Role of CSCs in Resistance to Cytotoxic Therapies: Chemo- and Radiotherapy

Cytotoxic anti-cancer therapies are mostly aimed at inducing tumor cell death. These treatment regimens can involve both the combination of radiotherapy and chemotherapeutic drugs, such as platinum-based drugs, antimetabolites, or anthracyclines [46]. Some chemotherapeutics have the same mechanism of action as radiotherapies, that is by means of direct DNA damage. Other chemotherapies, such as mitotic spindle poisons, inhibit cell division via their toxic effects on the dynamics of microtubules. Apart from the invasive off-target effects, chemo- and radiotherapies are associated with a mild, albeit not durable, response [46] The abscopal effect of radiotherapy when used in combination with immunotherapy is promising [47]. However, resistance to the currently used treatment strategies has been linked to CSCs and is considered as one of the main possible causes of poor results for CC and other malignancies [48,49,50]. Thus, through the understanding of the underlying mechanisms and oncogenic drivers by which the CSCs escape the radio- and chemotherapy, more effective treatments can be developed which could improve the clinical outcomes of patients with CC [51]. The intrinsic and extrinsic mechanisms of therapy resistance in CSCs have been extensively studied, and potential clinical use of CSC-targeting agents have been investigated in various cancers [45]. A transcriptome analysis of CCSCs from responder and non-responder groups to chemoradiotherapy identified several differentially expressed genes, including ILF2, RBM22P2, ACO16722.1, AL360175.1, and AC092354.1 [52].

We briefly report the main mechanisms by which CSCs contribute to resistance to anti-cancer therapies and the potential approaches to overcome this resistance (Table 2). This is followed by a description of the novel therapeutic strategies targeting CCSCs.

## 5. Resistance to DNA Damage-Induced Cell Death

Genotoxic agents and radiation treatment trigger the DNA damage response in which sensitive cancer cells fall into cell cycle blockade followed by induction of apoptosis. DNA damage sensor proteins, such as ataxia telangiectasia mutated-RAD3-related (ATR) kinases and ataxia telangiectasia mutated (ATM), are involved in these pathways [94,95,96,97]. Upon DNA damage, ATR and ATM kinases form complexes with breast cancer 1 (BRCA1) and poly ADP-ribose polymerase (PARP-1) to phosphorylate checkpoint kinase 1 (CHK1) and CHK2. These in turn activate targeted proteins and induce DNA repair [45]. CSCs can be resistant to DNA damage-induced cell death by promoting DNA repair capability through ATM and CHK1/CHK2 phosphorylation or by activating anti-apoptotic signaling pathways, such as WNT/β-catenin, PI3K/Akt, and Notch signaling pathways [45,46]. The c-MYC-CHK1/CHK2 axis regulates the DNA damage-checkpoint response, resulting in radiotherapy resistance in CSCs [98], while pharmacological inhibition of the CHK1 and CHK2 has been shown to sensitize CSCs to chemotherapy and/or radiotherapy [53,54]. In CC, aldehyde dehydrogenase (ALDH)-1 positive cells lead to radio-resistance by increasing DNA repair capacity and through preferential activation of the DNA damage checkpoint response [44]. As described further on, ALDH is a cytosolic enzyme responsible for the oxidation of intracellular aldehydes protecting cells from the potentially toxic effects of elevated levels of reactive oxygen species (ROS). However, the development of treatments that prevent DNA repair in cancer cells is more difficult than expected.

Chemo- and radiotherapy can induce expression of these DNA damage-checkpoint response pathways in non-CSCs and consequently activate cellular stress response and enhance stemness characteristics. Therefore, non-CSCs are more able to survive selectively. Thus, chemo- and radiotherapy can lead to an accumulation of a CSC subpopulation with higher innate resistance to these same therapies [11]. Many different approaches targeting CSC pathways and anti-apoptotic Bcl-2 family proteins are currently under clinical evaluation [11,45,46,55,56]. Most data about the role of PARP inhibitors (PARPi) in gynecologic malignancies specifically involve ovarian cancer. However, the role of PARPi in the treatment of CC is also currently being studied [57].

## 6. CSCs’ Quiescence

In addition to a robust DNA damage response, CSCs also undergo a persistent quiescence state which may contribute to therapy resistance. This is because some of the cytotoxic agents only target cancer cells that are highly proliferating [99,100,101,102]. Once treatment stops, these quiescent CSCs can re-enter the cell cycle and activate cell growth and proliferative signaling pathways, thus accelerating tumor regeneration [101]. The patterns of recurrence and acquired resistance that are observed in post-therapy cancer patients can be explained by the quiescence of CSCs. A deeper understanding of the mechanisms involved, whether activated or silenced, could prove useful for employing combinatorial therapeutic strategies to manipulate and sensitize CSCs to chemotherapy [101]. Activated TGF-β signaling, which is involved in triggering cytostatic signals, can lead to cisplatin resistance by driving the dormancy of CSCs in mouse squamous cell carcinoma [103]. Likewise, a subpopulation of CSCs undergoing epithelial-mesenchymal transition (EMT) has a slow rate of proliferation, thus conferring resistance to anti-proliferative drugs in breast and skin cancer models [104]. To overcome this resistance, three distinct approaches to therapeutic interventions have been put forward. These are allowing cells to remain dormant indefinitely, reactivating dormant cells, and increasing their sensitivity to anti-proliferative drugs, and/or eradicating dormant cells [58]. Relevant molecular mechanisms involving either the maintenance of quiescence indefinitely or eliminating this cancer cell subpopulation have been investigated with the aim of developing pharmacological interventions [58]. For instance, ablation of cytosolic phospholipase A2alpha (cPLA2α), which is a key mediator of pathophysiology of cancer and inflammation, has been shown to markedly improve chemosensitivity in CC via suppressing β-catenin signaling [59].

## 7. Epithelial-Mesenchymal Transition (EMT)

EMT and stem cell markers are co-expressed in circulating tumor cells from patients with metastatic lesions [105]. EMT induction or activation of EMT transcription factors, such as SLUG, SNAIL, SLUG, TWIST, and ZEB1/2, can confer stem-like features in cancer cells [106]. In addition, EMT activation is associated with therapeutic resistance by inducing cancer cells to exhibit stem cell-like characteristics, which promote invasion of surrounding tissues and the underlying drug resistance [60,61]. However, in most cases, the molecular mechanisms responsible for EMT and the resulting resistance are not clear. Cells undergoing EMT may stop dividing and enter a state of quiescence [60,61] and therefore, circumvent most conventional treatments which target actively dividing cells [107]. For instance, in oral cancer, cells with a SNAIL-mediated EMT phenotype exhibit quiescence and are highly resistant to chemotherapy [108]. The onset of EMT in CC increases the CSC subpopulation, increasing the metastatic potential of CC and leading to chemoresistance and radio-resistance. Hence, inhibiting EMT in CC cells sensitizes them to drugs and radiation [62]. Several molecular mechanisms responsible for maintaining the constitutive activation of the EMT pathway are being investigated [62]. These may prove clinically useful for developing new prognostic biomarkers and therapeutic targets for CC invasion and metastasis.

## 8. Tumor Microenvironment

The tumor microenvironment (TME) consists of tumor cells, tumor stromal cells (including stromal fibroblasts), endothelial cells, immune cells (such as macrophages, microglia, and lymphocytes), as well as the non-cellular components of extracellular matrix [109]. The concept of TME originated from studies showing that tumorigenesis preferentially specializes in niches within healthy tissue and in premetastatic niches [110]. TME is created by the influence of both the secreted factors from the primary cancer and from the host cells. This enhances the dissemination and survival of CSCs. The TME plays a crucial role in the establishment of a CSC niche via the provision of a quiescence promoting niche and by enhancing tissue invasion. In the primary tumor, a CSC niche is established in an anatomical location which provides more nutrients and signaling gradients, as it is usually residing close to a highly vascularized bed [110]. In some highly angiogenetic cancers, including advanced cervical cancer, CSCs can also cross lineage-differentiation boundaries to form different types of vascular cells [111,112,113]. CSCs can differentiate into functional endothelial cells to form blood vessels (vascular mimicry) [114].

Vasculature and blood-vessel-derived angiocrine factors are key components of TME. CSCs express angiogenic factors to promote vascular growth and enhance tumor growth. Simultaneously, endothelial cells create vascular niches through angiocrine-signaling to regulate CSC behavior, thus providing a chemoprotective microenvironment for CSCs and metastatic tumor cells. In addition, alterations in the vascular microenvironment can reactivate the dormant disseminated tumor cells, leading to relapse. In conjunction with traditional chemotherapy, there is evidence that treatments that disrupt angiocrine crosstalk can chemosensitize otherwise chemoresistant CSCs and improve treatment efficacy [115].

The TME and its CSC niche are very likely to be different in each tumor type [116]. In the cervix, the squamocolumnar junction, also known as the transformation zone, is considered a baseline lymphangiogenic niche in the cervical tumorigenesis [117].

Stimuli from the CSC niche may be another route for treatment resistance. The CSC microenvironment could create an imbalance between CSC differentiation and self-renewal [118] by stimulating signaling pathways, such as Notch and Wnt. These in turn may facilitate evasion of CSCs metastasis and anoikis, thus altering divisional dynamics, and facilitating repopulation through symmetric division [61,63]. For instance, specialized microenvironments of bone marrow endothelial cells are important for homing and engraftment of both normal HSCs and leukemic cells [119]. In AML, extracellular matrix components and signaling molecules in the HSC microenvironment promote cell survival, providing resistance to chemotherapy. In the case of glioma, it has been shown that intrinsic properties of glioma stem cells are very tightly regulated by specific signals derived from the niches, which help to maintain their undifferentiated state as well as their number [119]. Moreover, relationships between CSCs and their niches can be bi-directional. Apart from exploiting pre-existing microenvironments, glioma stem cells are also actively involved in shaping and generating their niches via intricate crosstalk with diverse components of both surrounding and distant tissues [119]. It has been shown that components of CSC niches may be significantly related to the metastatic potential of CSCs. For example, VEGF Receptor 1 (VEGFR1) signaling from distant primary tumors induces MMP9 in clusters of pre-metastatic lung endothelial cells [119]. In addition, integrins and adhesion molecules may be associated with migration of CSCs [117].

With increasing evidence supporting the important role of the TME in enhancing CSC-mediated tumor propagation, indirect targeting of CSCs may occur via components of the TME, such as cancer-associated fibroblasts (CAFs) or tumor-associated macrophages (TAMs), that secrete factors that induce EMT [61,63]. Similar to what happens in hepatocellular carcinoma, in CC, miRNA125 delivered via TAM exosomes may significantly suppress the CSC phenotype, thus limiting drug resistance [64,120].

## 9. Hypoxia

Pre-clinical investigations clearly demonstrate that hypoxic microenvironments in solid tumors significantly impair tumor response to anti-cancer therapies (radio-, chemo-, and immunotherapy), increase cancer aggression, and promote progression and metastasis [65]. Indeed, hypoxia is considered as an independent predictor of disease progression, treatment failures, and higher metastatic potential in many cancers, including CC, sarcoma, breast, and prostate cancer [66].

Hypoxia activates several signaling pathways by inducing hypoxia-inducible factors 1α and 2α (HIF1α, HIF2α) or phosphatidylinositol 3-kinase (PI3K/AKT), which bind to promoters containing the hypoxia-response element (HRE). This in turn, promotes tumor survival via the upregulation of the expression of multiple genes associated with angiogenesis, apoptosis, metabolic regulation, and pH balance. Activation of the PI3K/ATK pathway promotes CSCs by activating HIF1α and HIF2α as a feedback loop, and this cascade leads to the induction of stemness and self-renewal [121]. In ovarian cancer cells, HIF induces stem cell properties, promoting ovarian CSCs adaptive stress response and resistance to therapy [67]. Hypoxia enhances the radioresistant phenotype of ALDH-1-positive CSC-like cells from the CC lines, HeLa and SiHa, by improving post-radiation DNA repair and preferentially activating the DNA damage checkpoint response [44].

Low-oxygen conditions maintain CSCs in a quiescent state with a low proliferation rate, thus enhancing chemo- and radio-resistance [67]. A higher local concentration of oxygen improves the efficacy of radiotherapy [122]. Thus, oxygen therapeutics by tumor oxygenation has been utilized as radiosensitizers with encouraging results in improving patients’ responses to radiotherapy [65,66]. In CC, treatment with radiotherapy and hyperbaric oxygen showed significant improvement both in local cancer control and patient survival [68]. In later studies involving CC patients who were treated with radiotherapy or surgery, all patients with tumor pO2 values less than10 mm Hg had a lower overall and disease-free survival than patients whose lesions were better oxygenated [69,70]. This supports a role for hypoxia in radio-resistance and increased tumor aggressiveness.

## 10. Multidrug Resistance (MDR) and ALDH-Associated Resistance

Side population cells exhibiting a cancer stem cell-like phenotype have been detected in a variety of different solid tumors, including CC [61,123]. These side population cells show increased expression of drug-transporter proteins, including MDR1 (ABCB1), ABCC1 (MRP1), and ABCG2. The overexpression of ABC protein is one of the main protective mechanisms for CSCs in response to chemotherapeutic agents [124]. This facilitates the expulsion of Hoechst dye and, more importantly, cytotoxic drugs, leading to higher resistance to chemotherapeutic agents and disease relapse [125,126,127]. In ovarian cancer, CSCs are sensitive to drugs, such as fumitremorgin C and verapamil, that block ABC transporters [71]. The use of ABC transporter inhibitors in combination with chemotherapy is currently undergoing pre-clinical investigation in CC and other cancers [72,73,74].

As previously mentioned, ALDH protects cells from the potentially toxic effects of raised ROS levels. High ALDH levels are present in both normal and CSCs and have been shown to be involved in chemo-and radiotherapy resistance [128,129,130]. ALDH activity has been shown to be a potential selective marker for CSCs in CC and other different types of gynecological cancer [75,76,77,78,79,80,81]. Several general and isoform-specific ALDH inhibitors have been shown to be effective in pre-clinical models of gynecologic malignancies, supporting further clinical testing [82]. ALDH inhibitors, such as CM307 and. 673A, synergize with chemotherapy to reduce tumor growth. Thus, ALDH-targeted therapies hold promise for improving patient outcomes in CC and other gynecologic malignancies [82,83].

## 11. Epigenetic Programming

Epigenetic programming, involving DNA methylation, histone acetylation, microRNA (miRNA) expression, and chromatin remodeling, is implicated in causing cancer cells to regain stem CSC-specific features [131,132]. Dysregulation of epigenetic mechanisms can contribute to the progression of CSCs due to abnormal epigenetic memory. Consequently, agents targeting epigenetic programming may be potential anti-CSC therapies [133,134]. DNA methyltransferase (DNMT) inhibitors are a class of anti-CSC compounds, which are already being used as part of the management of different types of malignancies, including CC [84,85]. Histone deacetylases (HDACs) are chromatin-remodeling enzymes involved in histone acetylation, which can modulate chemotherapeutic resistance in CC and various other cancers [86,87].

miRNAs are important regulators of gene expression by inducing mRNA degradation and/or translational repression via interaction with the 3′ untranslated region (3′-UTR) of target mRNAs. miRNA influence gene translation in both canonical and non-canonical ways [135]. In some cases, miRNAs have been shown to interact with different regions on genes, including promoters, and are involved in the activation and regulation of gene transcription [136]. It has been revealed that miRNAs have an essential function in the biology of CSCs via the regulation of signaling pathways of stemness, EMT, differentiation, and carcinogenesis in the cells [137]. Abnormal miRNA expression can lead to tumor suppression or act as an oncogene in various cancers [138]. The miR-302–367 cluster was identified in the tumor-initiating glioma cells, and it has been shown to suppress the growth of CCSCs via the negative regulation of the cyclin D1 and the AKT1 pathway. This suggests that the miR-302–367 cluster may serve as a potential therapeutic reagent even in CC [91]. In addition, in the CC cell lines, Hela, Siha, CaSki, and C33A, miR-23b suppresses stem marker expression, decreases the size and amount of the tumorsphere and decreases cell resistance to cisplatin via inhibition of the expression of aldehyde dehydrogenase 1 family member A1 (ALDHA1) [92]. In addition, in CC patients miR-145 induces CSC differentiation and reduces cell invasion and colony formation, as well as displaying a positive correlation to survival. Indeed, when nude mice were injected with adenovirus carrying miR-145, there was a significant reduction in tumor growth, leading to increased survival [93].

To date, there is evidence to support the testing of novel combinatorial therapeutic strategies based on administering drugs commonly used in clinical practice and epigenetic regulators (such as DNMT inhibitors, HDAC inhibitors, or miRNAs) to improve therapeutic efficacy in solid cancer patients and overcome the limitations of chemotherapy alone [85,88,89,90]. Since epigenetic mechanisms are key regulators of CSCs, standard drug combinations together with new epigenetic-type agents that target and kill CSCs in CC, without adversely affecting normal stem cells and consequent adverse toxicity in cancer patients, hold great promise in oncology.

## 12. Anti-CCSC Therapeutic Strategies

The selective targeting of CSCs is a promising therapeutic strategy aimed at eliminating the cancer development and minimizing recurrence [139]. Many therapeutic agents have emerged against CSCs and have been evaluated in pre-clinical cancer models and in clinical trials [140]. The success of suppressing chemotherapy resistance of CSCs by anti-CSC agents relies on the identification of molecular pathways, miRNAs, and niches that selectively regulate CSC function [141]. However, to date, although certain chemicals, such as molecular iodine, apigenin, doxycycline, morusin, phenethyl isothiocyanate, zolendronic acid, and A1E (which is derived from 11 oriental medicinal plants) have been effective in treating CCSCs [9], there is very limited development of specific drugs and/or molecules targeting CCSCs [142].

Currently, several research groups are attempting to identify new target genes, proteins, and signaling pathways that are involved in the stemness of CC cells. CSC-specific markers, such as CD133 and CD49f, and signaling pathways, including Hedgehog, PI3K/Akt/mTOR, Wnt, or Notch, have been largely used as therapeutic targets [25]. A dual-targeting strategy, consisting of outer layers and inner parts, has been proposed to target CSCs [143]. Anti-tumor therapeutic agents are loaded on the outer layer targeting cancer cells, whereas antibody-drug conjugates (ADCs) targeting CSC are encapsulated in the inner parts. This system can accumulate and become concentrated in tumor tissue through enhanced permeability and retention. The drugs on the outer layer are meant to kill the cancer cells, and then ADCs release gradually from the inner part at the tumor site targeting CSCs and thus, help to eliminate cancer cells. In this way, the CSCs may be killed using these therapeutic drug combinations, and the beneficial effect of chemotherapy in cancer treatments could improve [143]. However, no studies have yet reported on the use of dual targeting to treat CCSCs. Therefore, more studies on dual targeting are needed specifically in CC.

CSC targeting with nanoparticles (NPs) is another potentially effective therapeutic approach [13]. NP-enabled therapies have been designed to target stem cell-specific signaling pathways and thus, inhibit stem cell-related functions. In particular, NP-mediated photothermal therapy has been shown to be effective for both breast CSCs and cancer cells [144]. CCSC-targeted salinomycin NPs provide a potential selective target that can efficiently eradicate CCSCs [145]. However, there is poor bioavailability and serious side effects, limiting their clinical application. To overcome these limitations, Sal-Docetaxel-loaded gelatinase-stimuli nanoparticles could be a promising strategy to enhance anti-tumor efficacy and reduce side effects by simultaneously suppressing CCSCs and non-CCSCs [146].

However, NP-enabled therapies are still far from an ideal CSC-specific targeting therapy, especially because specific, sensitive markers or an equal combination of different markers, and distinctive CSC-signaling pathways have not yet been characterized for each CC type. Promising therapeutic strategies based on CSC targeting have been described, such as the targeting of CSC-signaling pathways, CSC niche, and CSC mitochondria, with advances being made in CSC-targeted drug delivery systems (DDSs) [147].

Despite more studies having been carried out using CSC-targeting therapies, there are still several limitations which are not easy to overcome. CSCs are typically present in very low numbers in tumors, accounting for approximately 0.1–10% of tumor cells [148]. Furthermore, CSC-targeted therapy may damage normal stem/progenitor cells and block the regeneration of normal tissues, leading to tissue and/or organ dysfunction [13].

## 13. Conclusions and Future Perspectives

Numerous studies have been carried out to help in further understanding the molecular pathogenesis of CC and the progression of viral infections leading to this tumor. While most investigations have attempted to prove that the cause of CC is HPV infection, recent studies have aimed to determine the underlying factors and changes happening at the molecular level which are involved in the development and stemness of CC. There is now mounting evidence that CCSCs play a fundamental and strategic key role in cancer development and regression, including resistance to therapy.

The current conventional chemo- and radiotherapies target the differentiated cancer cells; and thus, CSCs are not harmed due to resistance to therapy. The main mechanisms whereby CSCs contribute to the resistance to anti-cancer therapies, together with the approaches to overcome this resistance, have been outlined in this review. Innovative treatment approaches for the elimination of CSCs in CC have been reported. The targeting of various stem-cell-related markers and signaling pathways has the potential to be a novel strategy for CSC-targeted therapy, such as through dual targeting and NP-enabled therapies. However, challenges for CSC-targeted therapy remain to be overcome, including potential damage to normal stem/progenitor cells. Therefore, further in-depth knowledge of the biology, function, and clinical implications of CSCs in CC therapy is crucial to develop more effective therapeutic modalities for patients with CC.

## Figures and Tables

**Figure 1 ijms-23-05167-f001:**
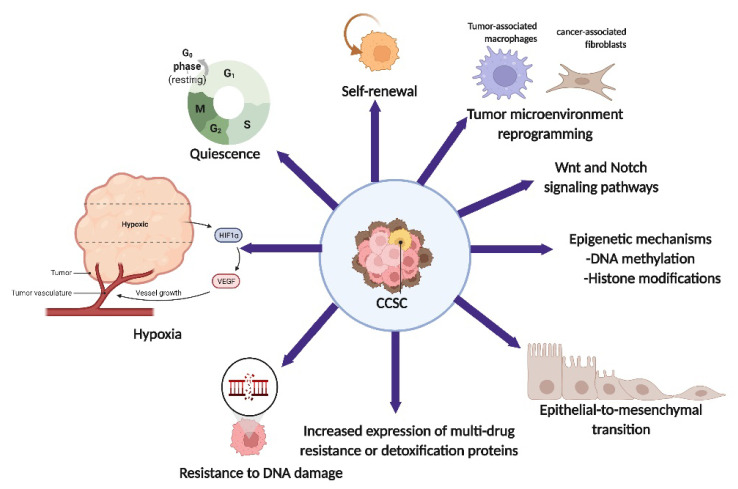
Illustration summarizing various mechanisms in cervical cancer stem cell (CCSC) contributing to chemoresistance. CSCs can contribute to chemoresistance through various mechanisms including quiescence, self-renewal, tumor microenvironment reprogramming, signaling pathways, epigenetic mechanisms, epithelial-to-mesenchymal transition (EMT), enhanced expression of multi-drug resistance or detoxification proteins, resistance to DNA damage, and hypoxia.

**Table 1 ijms-23-05167-t001:** A selection of published studies on CSCs in human cervical cancer.

Study	Sample	CSC Marker(s) and/or Phenotype(s)	CSC Characteristics and/or Clinical Significance
Feng et al., 2009 [26]	Primary tumor sphere culture	CD44^+^/CK17^+^	Chemoresistance; tumorigenicity
Bortolomai et al., 2010 [27]	3 cell lines; xenografts	ALDH^high^; SP	Sphere formation; tumorigenicity
López et al., 2012 [28]	4 cell lines; xenografts	CD49f^+^	Sphere formation; radioresistance; tumorigenicity
Zhang et al., 2012 [29]	HeLa cells; xenografts	SP	Increased invasiveness; tumorigenicity
Wang et al., 2013 [30]	HeLa cells; xenografts	SP	Colony formation; radio- and chemo-resistance; tumorigenicity
Liu & Zheng, 2013 [31]	4 cell lines and 5 primary tumor xenografts	ALDH^high^	Chemoresistance; tumorigenicity
Qi et al., 2014 [32]	HeLa cells; xenografts	SP	Radio- and chemo-resistance; tumorigenicity
Wang et al., 2014 [33]	HeLa cells	OCT4, SOX2 and ALDH	Colony formation; sphere formation; chemoresistance
Villanueva-Toledo et al., 2014 [34]	3 cell lines	SP	Colony formation; sphere formation
Liu et al., 2014 [35]	2 cell lines; xenografts	SOX2	Sphere formation; tumorigenicity
Kumazawa et al., 2014 [36]	HeLa cells; xenografts	CXCR4, Oct3/4, CD133, and SOX2	Sphere formation; radioresistance; tumorigenicity
Hou et al., 2015 [37]	179 tissue specimens	MSI1, ALDH1, SOX2 and CD49f	High expression of MSI1, ALDH1, and SOX2, and low expression of CD49f predict poor prognosis inspite ofpostoperative chemotherapy
Liu et al., 2016 [38]	SiHa cells; xenografts	CD44^+^/CD24^+^	Sphere formation; radioresistance; tumorigenicity
Ortiz-Sánchez et al., 2016 [39]	4 cell lines; xenografts	CK-17^+^, p63^+^, CD49f^+^, ALDH^high^	Sphere formation; tumorigenicity
Xie et al., 2016 [40]	52 tumor samples	ALDH1	ALDH1 expression predicts chemoresistance and poor clinical outcomes in patients with LACC receiving NAC prior to radical hysterectomy
Wei et al., 2017 [41]	Primary cell cultures	SP	Colony formation; tumorigenicity
Javed et al., 2018 [42]	Primary cell cultures	CD133^+^	Sphere formation; EMT and radioresistance
Li et al., 2019 [43]	6 cell lines; xenografts; 233 tissue specimens	NUSAP1	Sphere formation; EMT and tumorigenicity. High expression of NUSAP1 positively correlated with lymph node metastasis. Patients with high NUSAP1 expression have shorter 5-year metastasis-free survival
Yao et al, 2020 [44]	2 cell lines; xenografts	ALDH^high^	Sphere formation; radioresistance; tumorigenicity

Abbreviations. SP: Side population; NAC: neoadjuvant chemotherapy; LACC: locally advanced cervical cancer; EMT: epithelial-mesenchymal transition.

**Table 2 ijms-23-05167-t002:** Main cellular mechanisms of resistance of CSCs to therapies and potential therapeutic approaches.

Cellular Mechanism	Cancer Therapeutic Resistance	Therapeutic Approach	Example of Therapeutic Approach to CC
High DNA repair capacity and activation of anti-apoptotic pathways	Chemo- and radioresistance	Inhibition of the DNA damage checkpoints CHK1 and CHK2; targeting self-renewal and survival-related pathways (e.g. WNT/β-catenin, Hedgehog, Notch and PI3K/AKT/mTOR pathways); anti-apoptotic Bcl-2 family proteins; PARP family of enzymes [11,45,46,53,54,55,56]	PARP inhibitors (e.g. veliparib, olaparib, niraparib and rucaparib) are currently being studied [57]
Cell quiescence	Chemo- and radioresistance	Allowing cells to remain dormant indefinitely; reactivating dormant cells; eradicating dormant cells [58]	Inhibition of cytosolic phospholipase A2 alpha (cPLA2α) with efipladib improves chemosensitivity [59]
EMT	Chemo- and radioresistance	Targeting factors (e.g. cytokines, proteins, miRNAs, transcription factors, miRNA) and signaling pathways involved in EMT [60,61]	Plant products (e.g. anthocyanins, morusin and curcumin) inhibit EMT [62]
Tumor environment	Chemo- and radioresistance	Targeting the components of the tumor microenvironment (e.g. CAFs or TAMs) [61,63]	The upregulation of miR-125a sensitized to paclitaxel and cisplatin [64]
Hypoxia	Chemo- and radioresistance	Tumor oxygenation and oxygen therapeutics [65,66,67]	Hyperbaric oxygen and radiotherapy [68,69,70]
Multidrug resistance (MDR)	Chemoresistance	Inhibiting ABC transporters [71,72,73]	Stemofoline increases chemosensitivity by inhibiting P-glycoprotein [74]
ALDH-associated resistance	Chemo- and radioresistance	Inhibiting ALDHs [75,76,77,78,79,80,81,82]	Disulfiram-loaded vaginal ring potentially used for the localised treatment of CC [83]
Epigenetic Programming (e.g. epigenetic mechanisms, abnormal expression of miRNAs)	Chemo- and radioresistance	Inhibiting DNMTs and HDACs; manipulating miRNAs [84,85,86,87,88,89,90]	SGI-1027, a DNMT1 inhibitor, impairs CC cell propagation [85]. HDAC inhibitors (e.g. vorinostat, valproic acid, oxamflatin, 2-Oxo-1,3-thiazolidine, etc) may add to the efficiency of CSC therapy [86,87]. The miR-302–367 cluster [91], miR-23b [92] and miR-145 may serve as potential therapeutic reagents [93]

## Data Availability

This review paper does not report any new data.

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
