# Peer review of "Cancer Stem Cells and Their Possible Implications in Cervical Cancer: A Short Review"

_ijms, 2022, doi:10.3390/ijms23095167_

Round 1
Reviewer 1 Report
Overall, this is a written concise review on cancer stem cell and its potential as a therapeutic target for cancer. Although the title indicated that this review will focus on the translational potential of cervical cancer cancer stem cell, as written it appears that the content is a generic description of cancer stem cell and is not cervical cancer-specific. The author should consider change the title to fit the content or revise the content to focus on the advance of cervical cancer cancer stem cell and its translational potential.
Author Response
Authors' Response to Decision Letter (ijms-1693713)
Title: Cervical cancer and cancer stem cells: new translational opportunities?
Authors' Response
Dear Editors,
We are pleased to submit a revised version of the manuscript (ijms-1693713) entitled “Cervical cancer and cancer stem cells: new translational opportunities?”. We appreciate the reviewers’ judgment and have now modified the manuscript accordingly. We believe the manuscript has benefited as a result. The point-by-point reply to reviewers’ comments are shown below.
Reviewer 1
Overall, this is a written concise review on cancer stem cell and its potential as a therapeutic target for cancer. Although the title indicated that this review will focus on the translational potential of cervical cancer cancer stem cell, as written it appears that the content is a generic description of cancer stem cell and is not cervical cancer-specific. The author should consider change the title to fit the content or revise the content to focus on the advance of cervical cancer cancer stem cell and its translational potential.
Dear Reviewer 1, many thanks for the kind comments made on our manuscript, for the accurate revision and the useful suggestions.
In agreement with the reviewer, we have considered changing the title of the manuscript to “Cancer Stem Cells and their possible implications in Cervical Cancer: A Short Review”
Best Regards
Prof Jean Calleja-Agius
Reviewer 2 Report
This is an excellent work by the authors and I have no reservation in strongly recommending it for publication.
One aspect I found missing, for regulation of TME, what is the significance of cervical vasculature? ( blood and lymphatic) How does the vasculature contribute to CSC survival and renewal? However, the significance of vasculature is depicted in Figure 1.
Author Response
Authors' Response to Decision Letter (ijms-1693713)
Title: Cervical cancer and cancer stem cells: new translational opportunities?
Authors' Response
Dear Editors,
We are pleased to submit a revised version of the manuscript (ijms-1693713) entitled “Cervical cancer and cancer stem cells: new translational opportunities?”. We appreciate the reviewers’ judgment and have now modified the manuscript accordingly. We believe the manuscript has benefited as a result. The point-by-point reply to Reviewer’ s 2 comments are shown below.
Reviewer 2
This is an excellent work by the authors and I have no reservation in strongly recommending it for publication.
One aspect I found missing, for regulation of TME, what is the significance of cervical vasculature? (blood and lymphatic) How does the vasculature contribute to CSC survival and renewal? However, the significance of vasculature is depicted in Figure 1.
Dear Reviewer 2, many thanks for the kind comments made on our manuscript and for the useful suggestions.
The section “Tumor microenvironment” has been rewritten as follows:
The tumor microenvironment (TME) consists of tumor cells, tumor stromal cells (including stromal fibroblasts), endothelial cells, immune cells (like macrophages, microglia and lymphocytes), as well as the non-cellular components of extracellular matrix [78]. The concept of TME originated from studies showing that tumorigenesis preferentially specializes in niches within healthy tissue and in premetastatic niches [79]. TME is created by the influence of both the secreted factors from the primary cancer and from the host cells. This enhances the dissemination and survival of CSCs. The TME plays a crucial role in the establishment of a CSC niche via the provision of a quiescence promoting niche and also by enhancing tissue invasion. In the primary tumor, a CSC niche is established in an anatomical location which provides more nutrients and signaling gradients, as it is usually residing close to a highly vascularized bed [79]. In some highly angiogenetic cancers including advanced cervical cancer, CSCs can also cross lineage-differentiation boundaries to form different types of vascular cells [80-82]. CSCs can differentiate into functional endothelial cells to form blood vessels (vascular mimicry) [83].
Vasculature and blood vessel derived angiocrine factors are key components of TME. CSCs express angiogenic factors in order to promote vascular growth and enhance tumor growth. Simultaneously, endothelial cells create vascular niches through angiocrine signaling to regulate CSC behavior, thus providing a chemoprotective microenvironment for CSCs and metastatic tumor cells. In addition, alterations in the vascular microenvironment can reactivate the dormant disseminated tumor cells, leading to relapse. In conjunction with traditional chemotherapy, there is evidence that treatments that disrupt angiocrine crosstalk can chemosensitize otherwise chemoresistant CSCs and improve treatment efficacy [84].
The TME and its CSC niche are very likely to be different in each tumor type [85]. In the cervix, the squamocolumnar junction, also known as transformation zone, is considered to be a baseline lymphangiogenic niche in the cervical tumorigenesis [86].
Stimuli from the CSC niche may be another route for treatment resistance. The CSC microenvironment could create an imbalance between CSC differentiation and self-renewal [87] by stimulating signaling pathways, such as Notch and Wnt. These in turn may facilitate evasion of CSCs metastasis and anoikis, thus alterating divisional dynamics, and facilitating repopulation through symmetric division [74,88]. For instance, specialized microenvironments of bone marrow endothelial cells are important for homing and engraftment of both normal HSCs and leukaemic cells [89]. In AML, extracellular matrix components and signaling molecules in the HSC microenvironment promote cell survival, providing resistance to chemotherapy. In the case of glioma, it has been shown that intrinsic properties of glioma stem cells are very tightly regulated by specific signals derived from the niches, which help to maintain their undifferentiated state as well as their number [89]. Moreover, relationships between CSCs and their niches can be bi-directional. Apart from exploiting pre-existing microenvironments, glioma stem cells are also actively involved in shaping and generation of their niches via intricate crosstalk with diverse components of both surrounding and distant tissues [89]. It has been shown that components of CSC niches may be significantly related to metastatic potential of CSCs. For example, VEGF Receptor 1 (VEGFR1) signaling from distant primary tumors induces MMP9 in clusters of pre-metastatic lung endothelial cells [89]. In addition, integrins and adhesion molecules may be associated with migration of CSCs [86].
With increasing evidence supporting the important role of the TME in enhancing CSC-mediated tumor propagation, indirect targeting of CSCs may occur via components of the TMEt such as cancer-associated fibroblasts (CAFs) or tumor-associated macrophages (TAMs) that secrete factors that induce EMT [74,88]. Similar to what happens in hepatocellular carcinoma, in CC, miRNA125 delivered via TAM exosomes may significantly suppress the CSC phenotype, thus limiting drug resistance [90,91].
- We have also added the following references
- Baghban, R.; Roshangar, L.; Jahanban-Esfahlan, R.; Seidi, K.; Ebrahimi-Kalan, A.; Jaymand, M.; Kolahian, S.; Javaheri, T.; Zare, P. Tumor microenvironment complexity and therapeutic implications at a glance. Cell Commun Signal. 2020, 18, 59.
- Plaks, V.; Kong, N.; Werb, Z. The cancer stem cell niche: how essential is the niche in regulating stemness of tumor cells? Cell Stem Cell. 2015, 16, 225-238.
- Liu, Z.; Qi, L.; Li, Y.; Zhao, X.; Sun, B. VEGFR2 regulates endothelial differentiation of colon cancer cells. BMC Cancer. 2017, 17, 593.
- Shangguan W, Fan C, Chen X, Lu R, Liu Y, Li Y, Shang Y, Yin D, Zhang S, Huang Q, et al. Endothelium originated from colorectal cancer stem cells constitute cancer blood vessels. Cancer Sci. 2017, 108, 1357-1367.
- Eskander, R.N.; Tewari, K.S. Beyond angiogenesis blockade: targeted therapy for advanced cervical cancer. J Gynecol Oncol. 2014, 25, 249-259.
- Li, F.; Xu, J.; Liu, S. Cancer Stem Cells and Neovascularization. Cells. 2021, 10, 1070.
- Pasquier, J.; Ghiabi, P.; Chouchane, L.; Razzouk, K.; Rafii, S.; Rafii, A. Angiocrine endothelium: from physiology to cancer. J Transl Med. 2020, 18, 52.
- Najafi, M.; Farhood, B.; Mortezaee, K. Cancer stem cells (CSCs) in cancer progression and therapy. J Cell Physiol. 2019, 234, 8381-8395.
- Balsat, C.; Signolle, N.; Goffin, F.; Delbecque, K.; Plancoulaine, B.; Sauthier, P.; Samouëlian, V.; Béliard, A.; Munaut, C.; Foidart, J.M.; et al. Improved computer-assisted analysis of the global lymphatic network in human cervical tissues. Mod Pathol. 2014, 27, 887-898.
- Yao, T.; Lu, R.; Zhang, Y.; Zhang, Y.; Zhao, C.; Lin, R.; Lin, Z. Cervical cancer stem cells. Cell Prolif. 2015, 48, 611-625.
- Consequently, we revised table 2.
Best regards
Prof Jean Calleja-Agius
Round 2
Reviewer 1 Report
Previous concerns adequately addressed.